# Supporting Innovative Person-Centred Care in Financially Constrained Environments: The WE CARE Exploratory Health Laboratory Evaluation Strategy

**DOI:** 10.3390/ijerph17093050

**Published:** 2020-04-28

**Authors:** Helen M. Lloyd, Inger Ekman, Heather L. Rogers, Vítor Raposo, Paulo Melo, Valentina D. Marinkovic, Sandra C. Buttigieg, Einav Srulovici, Roman Andrzej Lewandowski, Nicky Britten

**Affiliations:** 1School of Psychology, University of Plymouth, Plymouth PL6 8BX, UK; 2Institute of Health and Care Sciences, Gothenburg University Centre for Person-Centred Care (GPCC), 405 30 Gothenburg, Sweden; inger.ekman@gu.se; 3Biocruces Bizkaia Health Research Institute, Barakaldo, 48903 Bizkaia, Spain; rogersheatherl@gmail.com; 4IKERBASQUE, Basque Foundation for Science, Bilbao, 48013 Bizkaia, Spain; 5Centre for Business and Economics Research (CeBER), Centre of Health Studies and Research of the University of Coimbra, Faculty of Economics, University of Coimbra, Av. Dr. Dias da Silva 165, 3004-512 Coimbra, Portugal; vraposo@fe.uc.pt; 6Centre for Business and Economics Research, Faculty of Economics, INESC Coimbra, University of Coimbra, Av. Dr. Dias da Silva 165, 3004-512 Coimbra, Portugal; pmelo@fe.uc.pt; 7Faculty of Pharmacy, Department of Social Pharmacy and Pharmaceutical Legislation, University of Belgrade, Vojvode Stepe 450, 11000 Belgrade, Serbia; valentina.marinkovic@pharmacy.bg.ac.rs; 8Department of Health Services Management, Faculty of Health Sciences, University of Malta, MSD2080 Msida, Malta; sandra.buttigieg@um.edu.mt; 9Department of Nursing, University of Haifa, Haifa 3498838, Israel; beinav@gmail.com; 10Faculty of Management, University of Social Sciences, 90-113 Lodz, Poland; roman.lewando@gmail.com; 11Institute of Health Research, University of Exeter Medical School, St Luke’s Campus, Exeter EX1 2LU, UK; n.britten@exeter.ac.uk

**Keywords:** We-CARE, person-centred care, patient-centred care, person-centred care, person centred care, evaluation, complex intervention, quality of care, cost containment, ethically grounded, evidence-based model

## Abstract

The COST CARES project aims to support healthcare cost containment and improve healthcare quality across Europe by developing the research and development necessary for person-centred care (PCC) and health promotion. This paper presents an overview evaluation strategy for testing ‘Exploratory Health Laboratories’ to deliver these aims. Our strategy is theory driven and evidence based, and developed through a multi-disciplinary and European-wide team. Specifically, we define the key approach and essential criteria necessary to evaluate initial testing, and on-going large-scale implementation with a core set of accompanying methods (metrics, models, and measurements). This paper also outlines the enabling mechanisms that support the development of the “Health Labs” towards innovative models of ethically grounded and evidenced-based PCC.

## 1. Introduction

The World Health Organisation defines Universal Health Care as that ‘which all citizens can access without incurring financial hardship’ [1]. Many nations fail to provide this as a basic human right, with half of the global population failing to access fundamental healthcare, and around 100 million people experiencing extreme hardship as a consequence of healthcare costs [2]. This is particularly true of nations with large income disparities and those without centrally funded health systems. In Europe and the UK, the situation is less stark, but the rising costs of healthcare are a threat to this basic tenet of post-war health policy [3,4,5].

Cost containment/saving initiatives in healthcare have been a consistent feature of health policy across Europe since the global economic crisis [6,7,8,9,10]. In the UK and Europe, a multitude of health policies have called for more efficiency and less waste [11], better integration of healthcare practices and services [12,13], the adoption of techniques of ‘lean’ management in the public sector similar to the ones utilized in the private sector [14,15], and, in some cases, privatization and competition [16]. Such policy drivers have also been in response to the changing nature of medicine and public health [14,15]. In some regions, people are living longer albeit more incapacitated lives, whilst in other regions, child mortality and early preventable death result from poverty and a marked social gradient of health [17,18]. Underlying these issues is the reality of healthcare spending, which is continuing to rise year after year as a growing percentage of GDP [19]. With EU healthcare spending estimated at €1.457 billion per year, saving just 1% could result in a saving of € 14.6 billion [20]. Healthcare, thus, holds great potential for cost containment.

The WE-CARE project (Grant Agreement 602131) was funded between September 2013 and August 2015 under the EU’s Seventh Framework Programme for Research (FP7). The project and its consortium were convened to tackle cost containment and quality care in recognition of the role these play in ensuring the accessibility and affordability of future health care for EU citizens. A key output—the WE-CARE Roadmap [21]—proposed seven interdependent themes to facilitate this aim. These themes consisted of: (1) Core drivers: Person-Centred care (PCC) and Health Promotion (HP); and (2) five critical enablers for their implementation: technology, quality measures, infrastructure, incentive systems, and contracting strategies.

Complex interventions that address cost containment and the quality of care through core drivers of PCC and HP are likely to require at least some critical enablers for successful implementation. These interventions will also require commitment from a range of key stakeholders, who must come together to find ethical and sustainable solutions [22]. A shift in thinking to one based on ‘collaborative action (CA)’ [23], which will include people, communities, health and social care providers, government at various levels, and a range of Non-Government Organisations (NGOs), is thus essential to establish these interventions. Guided by this approach, innovative models are required across Europe to prototype healthcare based on HP and PCC, emphasizing preventive self-assessment above reactive expert care. Such models must learn from each other and share some common elements, but may differ in geographical and health care system characteristics. Financed through different sources (e.g., EU funding for evaluation, government funding for implementation) their benefits may stretch far beyond the initial test beds.

### 1.1. Why Person-Centred Care (PCC) and Health Promotion (HP)?

PCC and HP are synergistic and a vital part of modern and ethical health care. A growing evidence base suggests that these core approaches can improve health outcomes [24], maintaining health care quality without increasing costs [25,26].

‘Person-centred care’, rather than ‘patient-centred care’, is emphasized because the word ‘patient’ is associated with a passivity in medical encounters [21,26]. For example, patients often adapt to the professional norms of healthcare organizations, rather than receiving care focused on their own resources and needs, preferences, and values [27]. Our notion of PCC is underpinned by the ethics of personhood and the capability approach [28]. The term ‘patient’ therefore is not commensurate with the person as an active partner in care and co-creator of their care [21,26]. Indeed, these two roles of the patient as a ‘person’ accurately describe the radical paradigm shift from patient- to person-centred healthcare that is necessary to reflect the philosophical and ethical changes in the delivery of PCC.

Health Promotion as an approach aims to inform, influence, and support people, communities, and organisations to improve health. Supporting people to increase control over their health is in essence health promoting, both for the individual and society [29,30]. HP activities can work in synchronicity with PCC if developed in partnership with the person, taking into consideration their life context and socioeconomic conditions [31,32].

### 1.2. Testing the Roadmap: A European Network for Cost Containment and Improved Quality of Health Care

COST Action 15222 ‘Cost Cares’ was funded by the EU Commission to create the impetus in both the research and development required to design and test innovative exploratory health laboratories (EHLs) to implement PCC and HP across the EU. This paper sets out a strategy for evaluating them.

## 2. Methods

### Defining and Understanding the Role of Critical Enablers

To understand how the EHLs might work to deliver PCC, HP, and cost outcomes, it was first necessary to develop Programme Theories (PTs). PTs describe how interventions (service, treatment, policy) are thought to work by specifying the ways in which they produce outcomes. They are a set of causal relationships often referred to as “If-Then” statements. They can also be written or represented graphically to show the relationships between cause and effect. PTs are also useful for understanding both the positive and negative impacts that can occur when interventions are implemented. They are often accompanied by logic models, which help plan and evaluate interventions based on their internal logic, and the role of context in supporting successful delivery and evidence acquisition. We created evidenced-based PTs to specify how EHLs would deliver PCC, HP, and cost outcomes through the critical enablers detailed in the We-CARE Roadmap (see Figure 1). Repeated here for clarity the critical enablers are (1) information technology (IT), which describes the use of computers or other computerized devises to store, transmit, and receive data to support PCC planning and care coordination, for handling and communicating health and evaluation data, and for delivering PCC and HP interventions. (2) Quality measures, such as organizational processes, that ensure health services increase the likelihood of the desired health outcomes consistent with current scientific knowledge, which take into consideration an individual’s preferences, and ensure that health services are effective, affordable and accessible to all citizens. (3) Infrastructure to create the necessary resources and structures that support the shift from health systems that are excessively hospital-centric and biomedically-oriented, to those which value continuity, responsiveness, and multidimensionality in community care, e.g., shifts in staffing, training, and delivery of care. (4) Incentive systems that reward PCC processes and outcomes, such as personal health goals, PCC plans, improvements in patient self-efficacy and experiences of care, and HP activities. This will require an expansion and critical revision of existing system-based biomedically driven performance indicators. (5) Contracting strategies that define and endorse PCC incentive systems and infrastructural support and efficiencies for EHLs, purchasing strategies and contracts between payers and providers of healthcare that promote the alignment in organisational goals based on PCC, HP, and cost containment. (6) Cultural change that represents shared assumptions, values, and beliefs that govern how people behave in an organisation. Receptiveness or readiness to change is considered a prerequisite for EHLs. As other critical enablers are modified within a given EHL, cultural change towards PCC, HP, and cost containment may present as either a pre-requisite and/or a natural consequence of development. The addition of this sixth critical enabler represents the importance of organizational culture in achieving PCC and cost stability.

Upon establishing agreed definitions of the above enablers, the next step was to hypothesize how these might work to support the aims of an EHL. Following this step, the literature was searched to detect evidence for the hypothesized statements, referred to as ‘if-then’ statements. To expedite this process, tables of ‘if-then’ statements were compiled, which, in keeping with the evaluation methods of critical realism [33,34], permitted the compilation of patterns of causal chains within the EHL. For example, IF condition X is in place (e.g., practitioners are incentivized to engage in shared decision making with patients), THEN outcome Y might follow (e.g., patients will feel like they are taking an active role in rehabilitation planning), thus improving service user experiences of PCC [35]. This task facilitated exploration of how the critical enablers interacted with PCC and HP to improve quality PCC and cost containment (see Figure 1). The points at which PCC and HP intersect with each of the critical enablers in Figure 1 are referred to as intersection points (e.g., PCC and information technology (IT)).

## 3. Evaluation Strategy

This section describes the considerations and necessary steps for evaluating and implementing EHLs to improve quality PCC and cost containment. First, the practice of PCC is explored, and then the role of critical enablers is illustrated. A number of controlled studies have been performed comparing PCC to usual care [21,36,37,38]. The core components in the interventions have been to listen carefully to the patient’s illness narrative and to mutually agree on a health plan. The true case story (see Figure 2) previously published in a position paper demonstrates how the patient narrative can open up and reveal information needed for the patient and the professionals to be able to agree on a relevant health plan [39]. This is concordant with the theory and philosophy that PCC is based on starting with each person’s capability and wish to take responsibility for their own health. The true case below is a vignette based on a real person to illustrate how PCC can be applied in practice through a worked example.

### 3.1. The Illness Story of Mr. G: An Example of the Enablers of PCC and HP

PCC for Mr. G was facilitated via various critical enablers, detailed in the following:

Information technology: The medical documentation and information (in the patient records) as well as the commonly formulated treatment and health plan are digitalized and accessible to Mr. G and his providers in a way that he comprehends and can agree or ask questions about. In formulating the health plan, Mr. G was supported by a digital patient decision aid [40]. Health information technology (IT) systems support the smooth flow of information between services, and to and from citizens and their families. Artificial intelligence might facilitate this and help improve interactions with patients [41]. Quality measures: Mr. G was invited to download an app after his first myocardial infarction where he can follow the development of symptoms and well-being and contact health care services for help and support with formulating his personal health plan. Contracting strategies, incentives, and infrastructure: The infrastructure supported cumulative documentation according to the criteria for PCC. This was linked to incentive payments for the whole team. This type of incentive payment includes quality measures (care plans) that are sanctioned and contracted between the provider and commissioner organisation.

### 3.2. Causal Relationships between Enablers, Outcomes, and Measurement

Program theories (PTs) are useful ways in which to facilitate an understanding of how complex interventions work; in this case, how the critical enablers could work with PCC and HP interventions to generate cost containment and quality PCC outcomes. Table 1 provides 11 worked examples of PTs referred to as ‘if-then’ statements with explanatory ‘because’ statements and associated suggestions for assessment or measurement. Instruments and methods to assess PTs should be carefully selected and the use of mixed methods is advised. Knowledge base and practical constraints will add to the existing complexity of measurement and evaluation. The type of design employed can help remedy some of these issues. For example, beginning with small-scale and qualitative assessment will help determine what to measure and how to measure it, and what improvements to expect. Ensuring measures or assessments capture professional and patient partnership work in care planning is key for emphasizing the importance of this for PCC.

The following PTs are presented here as examples of a larger body of work (available from the first author) conducted to inform the design and evaluation of EHLs. Table 1 presents seven different types (A–G) of evidence-based PTs that could shape the design of an EHL.

Type A *(*contracting strategies for quality and cost outcomes) PTs represent how contracting strategies could operate at macro and meso levels to support quality PCC and contain cost. In the two examples provided, ‘alliance’ or ‘partnership’ models contract to deliver an EHL based on shared or co-designed PCC and HP objectives to improve quality PCC and costs. This fosters trust and productivity based on collective ownership and the sharing of risk and reward within EHL. A mixture of quantitative and qualitative measures of delivery and management team dynamics, and progress towards aligned goals (e.g., PCC health plans), and costs over time could be used to ascertain the success of the contracting strategy. These enablers provide causal mechanisms for cost and quality outcomes at macro and meso levels within the EHL.

Type B *(*incentives and contracting strategies for quality PCC resulting in cultural change*)* PTs represent the potential for contracting strategies combined with incentives to improve cost and quality outcomes by providing incentives at multiple levels across the EHL. For example, if cost effectiveness is measured across the whole care chain with the savings provided to all participants, this creates the potential to act as an incentive towards aligned PCC and cost goals. To combat perceptions of unfairness in the equal distribution of savings across the system, objective measures of effort will need to be employed. These measures, however, should to be balanced against the knowledge of the operational context. For example, settings low on staff resources may seem to have contributed less towards the achievement of savings across the chain. Ensuring that contextual knowledge supports objective measurement will help communicate the conditions of contributors towards the savings gained and shared. Long-term planning and monitoring, active communication, and shared goals will help mitigate against perceptions of unfairness. Redistributing resources based on savings can help achieve the stated organization goals and thus improve the sector’s efforts where these are perceived to be lacking. These seemingly radical shifts align to the principles of fair division and social choice [59]. Over time, resultant cultural change across the system could be operationalized as permanent transformation of routines/habits. Measures of PCC and HP routines/habits, savings distribution, and measures of patient experience of care could help establish if this strategy is beneficial.

Type C *(*contracting strategies, incentives, and quality measures for cost and quality*)* PTs combine contracting strategies with incentives and quality measures to effect change in quality and costs. These build on type A and B PTs by, for example, suggesting that if contract payments are made at the same time to all providers and tied to measures of PCC and HP, this fosters trust and productivity by reducing the misalignment and unproductive competition between partners and reduces transaction hazards operating at macro and meso levels within the system.

Type D *(*incentives for quality PCC*)* PTs work at the micro level with incentives applied equally to all delivery staff irrespective of hierarchy or professional grouping [22,46] (e.g., patient feedback forms at clinic and ward levels). For quality PCC outcomes to be achievable, incentives must ensure that the reward system motivates individuals to align their own goals with those of the organization (EHL) [60,61]. As the PCC approach is based on qualitative changes, financial incentives may not be the best type of incentive to test. It has been long recognized that financial incentives are positively related to quantitative performance (e.g., number of tasks completed) but not necessarily with performance quality [62,63,64,65]. Thus, ideally, particularly since PCC is based in an Aristotelian ethics of virtues, the incentive systems should be a combination of financial and non-financial rewards (e.g., recognition, positive feedback from leaders, promotions, money, as well as target setting and performance evaluation itself) [66,67]. These rewards would be directed to all EHL members, since in “a complex network of interdependent relationships” [68,69] necessary for PCC implementation, it is difficult to identify an individual contribution. The success of micro-level incentives can be measured by carefully selected patient experience measures and focus groups.

Type E *(*incentives, quality measures for cost, and quality PCC*)* PTs work by combining incentives with quality measures at macro and meso levels. For example, if a PCC quality measure is linked to an EHL accounting system and able to deliver cost containment information resulting from PCC processes, then the measure itself becomes the incentive. Quality measures therefore act as both an aligned incentive and measurement of implementation. A pre- and post-comparison of costs associated with PCC quality processes analyzed against quality measure scores would provide an assessment of effectiveness. A benchmarking strategy against non-EHL settings may be an example of a measurement process being itself an incentive. It is important to note that the cost containment may not be immediate, as some costs may be incurred upfront and/or it may take time for outcomes to stabilize or become apparent. EHLs employing longitudinal designs can help to account for these potential delays.

Type F (Information Technology for quality*)* PTs provide examples of how IT has the potential to improve quality. These PTs work to support patient self-management through mobile technology, for example, through symptom monitoring or appointment reminders, to help people manage their own health [52]. They may also operate to support the adoption of PCC electronic health records and care plans, which provide teams with the tools to maintain and share PCC information. Measurement and evaluation of these mechanisms would be tailored to detect changes in patient self-management activities, team effectiveness, and resultant health system impact (e.g., reviews, appointments attended, etc.). In the current COVID-19 context, remote monitoring of patients, video-linked consultations, and e-health interventions could provide an exciting opportunity to test the delivery of person-centred care remotely, with the potential to calculate costs compared to previous standard practice [70].

Type G *(*Infrastructure for quality PCC*)* PTs provide examples of how components of an organization’s infrastructure could help result in quality care at meso and micro levels. At a meso level, if staff training is provided to enhance professional skills to support patient empowerment and enhance professional communication skills, this then has the potential to improve PCC delivery and experience of care. Furthermore, using patient-reported measures to shape care planning and use of the feedback from these measures to improve staff training has the potential to embed the patient voice in quality improvement practices and shape equitable person-centred relationships between professionals and patients [54]. A multitude of measures are available to measure these outcomes [27] and for use in care planning in this way. However, sampling care plans with patient-reported outcome measures (PROM) and interviews with professionals and patients would be insightful.

These examples of PTs are not comprehensive, but they illustrate how those developing EHLs can use these and other mechanisms to design their interventions and corresponding evaluation strategies. For further guidance on the use of evaluation metrics and measures, see p3c.org.uk.

### 3.3. Design the Evaluation and Fit with Service Change

The evaluation of EHLs should address questions that will enable commissioners of health services and delivery organizations to implement, sustain, and scale up the innovations. Key evaluation questions for the EHLs will include those that probe PCC processes, practices, and patient experiences of PCC care as markers of quality PCC. The health outcomes measured should be relevant to the patient and their family, health care provider, and other decision-makers. Key areas of interest in the implementation of PCC are changes in functional ability, experiences of care, self-efficacy, and cost. EHLs will also be informed by Wilson and Cleary’s [68] model for integrating concepts of biomedical outcomes and measures of health-related quality of life: (i) Biological and physiological factors, (ii) symptoms, (iii) functional status, (iv) general health perceptions, and (v) overall quality of life). Specific questions (see Figure 3) will also probe the mechanistic relationship between the critical enablers and PCC and HP. These are referred to as intersection points.

### 3.4. Economic Evaluation and Specific Questions for Cost Containment

Irrespective of the type of intervention, commissioners and policy makers require proof that the additional health care resources needed to make the procedure, service, or program available to those who could benefit from it are justified [71]. The purpose of economic evaluation is to inform such funding decisions. An economic evaluation deals with both inputs and outputs (costs and consequences) of alternative courses of action, and is concerned with choices and consideration of the costs and benefits at multiple levels. EHLs will therefore have to evaluate the main costs involved in the change of a healthcare system towards PCC and HP. Weinstein [72] identifies costs related to changes in the use of healthcare resources, changes in the use of non-healthcare resources, changes in the use of informal caregiver time, and changes in the use of patient time (for treatment). In a similar way, Drummond et al. [71] identifies health sector costs, other sector costs, patient/family costs, and productivity losses. Measurement within economic evaluation expands beyond the healthcare system under study.

According to Weinstein [72], direct health care costs include all types of resource use, including professional, family, volunteer, or patient time, as well as the costs of tests, drugs, supplies, healthcare personnel, and medical facilities. Non-direct health care costs include the additional costs related with the intervention, such as those for childcare (for a parent attending a treatment), the increase of costs required by a dietary prescription, and the costs of transportation to and from the clinic; they also include the time family or volunteers spend providing home care. Citizen time costs include the time a person spends seeking care or participating in or undergoing an intervention or treatment. Time costs also include travel and waiting times as well as the time receiving treatment. Productivity costs include (1) the costs associated with a lost or impaired ability to work or to engage in leisure activities due to morbidity and (2) lost economic productivity due to death.

The World Health Organization (WHO) recognizes quality health care in those organizations that have a high degree of professional excellence, with minimum risks, good health outcomes for patients, and efficient use of resources [1,73]. To promote the health of the population, the WHO recommends key objectives for continuous quality improvement in health care. These include the structuring of health services, the rational and efficient use of both human and financial resources, and the guarantee of professional competence to citizens in order to meet their needs. Measures or questions relating to quality are likely to overlap and complement those relevant for cost containment (see Figure 4).

### 3.5. Measures and Metrics to Answer the Questions: A Core Minimum Data Set

The evaluation of the EHLs must contain the most suitable measures and approaches to answer the questions. Quantifiable measures or questions can either be aggregated (single criterion analysis) or handled separately (multi-criteria analysis). Careful consideration of the combination of qualitative and quantitative approaches is advised, particularly since different health systems display different capabilities in this regard. In terms of minimum design standards, at least two data collection points—pre-and post-intervention/implementation—are recommended. This is the minimum standard advised. Should the availability of knowledge, skills, and resources be forthcoming, more complex experimental and implementation-focused designs could be undertaken upon careful consideration of the amount of preexisting evidence for PCC in that particular context or condition [74]. Ideally, monitoring and data collection will be continuous and with feedback to practice, with long follow-up periods to capture lasting changes in care delivery and outcomes. To account for the variance in EHLs, a core minimum data set from each site with three categories of data is recommended: Routinely collected audit data or similar (e.g., collected at country or hospital level); questionnaire data specifically collected for the EHL; and qualitative data to support implementation development. Examples of suitable measures, depending on the focus of the changes in the healthcare system that are implemented, are given in Table 1. As an EHL is scaled out in practice, it may be necessary to add new measurements to capture unanticipated and/or unintended changes.

### 3.6. Sampling and Timing of Data Collection

A metrics framework provides the structure for planning the sampling and timing of data collection during the evaluation of an EHL. It is likely that data could flow from different sources, e.g., routinely collected data and quantifiable data, surveys, and qualitative data. The PT will guide the sampling strategies for data collection, the timing of data collection, and the various units of analysis. Qualitative approaches will always necessitate careful sampling because they are resource and time intensive. In contrast, an EHL may decide on a questionnaire to measure the experiences of all those using a service to canvas a broad view. The trade-off between qualitative approaches and more structured approaches involves considerations of depth versus breadth; different sampling strategies are required for different forms of data.

As qualitative approaches are effective for determining “how and why” the EHL is working, it will be important to consider a range of perspectives. Sampling should therefore aim for diversity in terms of ethnicity, social and economic status, age, disability, and health conditions. Services may also decide to film or record care interactions for ongoing implementation and quality improvement activities, using purposeful samples or random selection. Convenience and pragmatism will also play a role in any sampling procedure, which is common in applied health care research and evaluation, where time and resources are limited.

The phasing of data collection will likely include baseline data and follow-up data to mirror the timeframes of the intervention. It might also be necessary (providing sufficient justification and acceptability from practitioners and patients) that focused data capture on a specific element of the delivery is added into the core set of measures at particular times. For example, if communication or shared decision-making was an improvement target, implementing a tool that specifically addresses this issue of relational care could be used as both the intervention and data collection [75].

### 3.7. Routinely Collected Data, Linked Data Sets and Matched Cohorts

The potential to link health and social care data to understand an individual’s pathway following exposure to an EHL will be determined by local ethical restrictions, data flow, and governance guidelines. Linked data sets (or even unified data sets) allow for a longitudinal exploration of the impact of the intervention on service utilization (costs) and health using time series analysis or similar [76]. Analysis will be more powerful if compared to a control cohort (tracked by a unique identifier following explicit consent) of people who are part of a health lab. The use of techniques, such as propensity scoring, to identify and match control groups of service users are particularly helpful for this type of evaluation and service development [77].

### 3.8. Analysis

The analysis plan should be informed by the PTs and shaped by the evaluation framework. In principle, three main stages of analysis are envisaged. The first stage will commence with univariate analysis to examine each variable or source of data (for example, acceptability of services as a measure of quality or use of care plans as a measure of IT) independently. This could explore the time trends in say routinely collected data and the statistical properties of the data, e.g., the distribution of the data. Parallel qualitative analysis could seek to surface emerging themes. In the second stage, for each EHL, the PT will be tested to check if it is functioning as expected. In the third stage, findings both within and across the EHLs will be compared to answer the higher-order questions about the relationships between the quality of care and cost containment.

Working to understand trends in the data and other potential factors influencing outcomes (i.e., closure of a community hospital, or lack of out-of-hours primary care) will be a necessary effort. Collaboration between academic and health science partners will facilitate a robust evaluation, linking efforts to capture patient experiences and outcomes with cost indicators. The ultimate result will be a more nuanced story of how the intervention is delivered, experienced, and the extent to which it is achieving change. In this regard, it is important to note that change may not be immediate. Even if change is achieved quickly, impact on outcomes may require longer-term follow-up, especially, for instance, to demonstrate the cost-benefit ratio.

### 3.9. Barriers to Implementation

To convince European societies and key decision-makers at a national and an EU level that the WE-CARE Roadmap is viable, reliable evidence from the EHLs based on robust evaluation and implementation is required. Many barriers and uncertainties may threaten the implementation of PCC. The first is the quality and accuracy of the PT that underpins the EHL model; whether it includes all key aspects needed to provide PCC, if it examines quality care and/or cost, and the extent to which it includes the enablers within the EHL. The model should also be appealing and promise significant benefits, in order to convince key stakeholders of the potential EHL. However, not only is the quality of the theoretical model important, the legitimacy and reliability of the person or organization presenting the model to its future users is also crucial [78]. The engagement of authoritative local leaders who endorse the model to a range of stakeholders will be important to achieve early on in the process. This is likely to affect stakeholders’ perception of its quality and validity [79], as well as its advantage over alternative solutions [80,81].

#### 3.9.1. Enablers at Multiple Levels

The EHLs will affect people, their families, health professionals, and employees throughout the organization, including managers. Thus, a bundle of incentives for different groups will probably be required. Varied incentives, not only financial, as pay-for-performance, but also prospects of increased external recognition or legitimacy for participant organizations should be considered. The title of “the best provider”, achieved by public benchmarking, could be an example. This requires accurate outcome measurement. Incentive bundles can apply to three enablers of WE-CARE Roadmap: Incentive systems, quality measures, and contracting strategies.

#### 3.9.2. Contradictory Effects

The case-mix systems that are used in many European countries to finance hospital care are motivating providers to admit more patients, because the more patients they serve, the higher their income. If a hospital or a hospital ward agrees to become an EHL, the issue of contradictory incentives is likely to arise and must be overcome. For example, if, by implementation of an innovative community care EHL, more patients are cared for in the community, then the hospital will not receive money from the payer for those patients. The fixed costs of the hospital will remain, creating a deficit in the hospital system. A risk-reward sharing framework between the hospital and community provider could agree to cover hospital losses over the course of the project, but provisions for who will pay the fixed costs afterwards would need to be considered. Involving key stakeholders from across the system will be important to provide strategies to overcome these conflicting issues.

There should also be a distinction made between the average and the marginal cost of in-patient care. For example, the costs of a hospital ward (e.g., general medicine) are unlikely to differ significantly between a 10- or 20-patient occupancy. This means, that even if a treatment of a group of patients was organized outside of a hospital and the hospital infrastructure remained unchanged, the cost savings would be meagre or illusory. If, after introduction of the innovative care system, the medical infrastructure seemed unnecessary, then EHL employees would need to be motivated to support the EHL to ensure sustainability.

#### 3.9.3. Organizational Climate and Culture

The extent to which the organizational climate is favorable for EHL implementation must also be considered [80]. The implementation climate is more evident and less stable than the organizational culture and is thus more susceptible to amendments. Policies, procedures, and reward systems are those incentives that may effectively affect the implementation climate [82]. The other fundamental ingredient of a positive implementation climate is the extent to which important actors perceive the current healthcare delivery model as intolerable or unsustainable and are motivated for change, defined as cultural readiness [78]. The proposed model of the EHL should be compatible with stakeholders’ own norms and values (culture), as well as with their priorities [78,83].

To maintain a positive climate for the implementation of EHLs, important indicators related to citizen health, well-being, quality, costs, and other important factors should be presented to stakeholders. Thus, both the climate for change and incentives and reward systems call for accurate, objective, and verifiable measures viable to reflect the real performance in pivotal areas. If measures do not meet these requirements, this could undermine implementation [84]. Measures must clearly communicate PCC goals and feedback to participants indicating the degree of goal achievement.

To support PCC implementation and address potential barriers, each PT should be linked to a strategy with its own resources. Resources include knowledge, time, money, training, and in some cases physical space. Especially important is the access to widely understandable and convincing information and knowledge about PCC implementation, specifically about new work processes for the staff and the nature of care provided to patients and their social environment. If resources are not available, this creates a further barrier to implementation that must be effectively managed.

Organizational change begins with changes in individual behavior, although as numerous studies have shown, this is complex and challenging [80]. Ensuring the main actors do not perceive implementation of EHLs as threats to their own interests is a critical issue to address. Subjective interests are, however, not often easy to identify. Powerful actors, in particular leaders who at multiple levels across the system represent the core activity of the PCC implementation, must include physicians, nurses, allied health and social care professionals, people, and their communities. In EHLs, leadership should be transformational and innovative to create teams working to develop a workplace that is person centred. This is a key factor in the delivery and sustainability of PCC [85]. If this is achieved, it will promote cultural change and the upskilling of existing employees. Having several key people within the organization take on this role will ensure leadership sustainability.

Although these groups should support every change to augmenting healthcare quality, such as PCC, in reality, however, explicit or latent resistance can be a common problem [86]. The medical and health professions are built on an ideology to protect and care for humanity over economic profitability and self-reward [87], but contradictions between altruism and professional self-interest have been established. The excessive self-interest of individual doctors or groups of physicians should be mitigated by professional self-regulation and self-control [88].

### 3.10. Continuous Improvement

Since large-scale testing is the ultimate aim, it is assumed that a significant number of enabler elements will be in place when an EHL begins. As suggested earlier, the EHL will be underpinned by the PT that describes how the central work processes and independent actions of actors should be coordinated to deliver high-quality PCC. To be effective, the model, once elaborated, will require continuous adjustment not only to local environment factors but also to external and internal uncertainties emerging over time in each setting. Thus, some feedback and regulatory mechanisms should be an integral part of the model. The development and improvement of EHLs will be facilitated by a commitment to formative learning in response to the feedback from the evaluation data (data-driven improvement). There is a long tradition of using these methods to improve practice, and good evidence to suggest benefit [89]. Learning will vary by organization and setting. However, it will usually require a “plan-do-study-act” (PDSA) cycle or a similar process [90]. This will typically involve action learning sets [91] using quality improvement methodology [92]. Action learning sets are particularly suited to iterative complex intervention development as they focus on learning from interactions, thus providing a mechanism to reflect and problem solve. These skills are particularly important for health and social care professionals who are being asked to work in a different way, where this is likely to be challenging.

### 3.11. Co-Design and Participatory Action for PCC

Emancipatory research designs have been a core feature of community development and strengths-based approaches in social care. Such approaches value the lived experience and partnership with patients and the public in developing and evaluating services [93,94,95,96]. Research approaches based on these principles have in the past been subject to much derision but are now becoming recognized as critical to citizen-relevant and humanistic healthcare planning and evaluation, and align well with the philosophy of PCC. The UK standards for patient and public involvement in the planning and evaluation of health and social care are supported by academic, research, and government policy. Involving patients and the public in the consultation and shaping of EHLs is a core and fundamental standard we advocate.

## 4. Conclusions

This paper laid out a comprehensive plan for the evaluation of Exploratory Health Laboratories that aims to improve the quality of health care in the EU whilst also containing costs. The plan was developed by members of the WE-CARE FP7-funded project and COST CARES COST Action 15222 from a range of academic and professional backgrounds and different countries. This process identified PCC and HP as the solution, along with critical enablers to facilitate implementation. Examination of the intersections among and between these enablers, as well as the impact on quality of care and cost of care, via evidence-based PTs provides the justification for the design and incorporation of particular components into an EHL. Furthermore, the paper also described how these components and EHLs might be evaluated as complex interventions at micro, meso, and macro levels. This work and the resources it produced (www.COSTCARES.eu) are intended to serve as a reference material for those considering setting up EHLs or similar initiatives beyond the scope of this COST Action.

## Figures and Tables

**Figure 1 ijerph-17-03050-f001:**
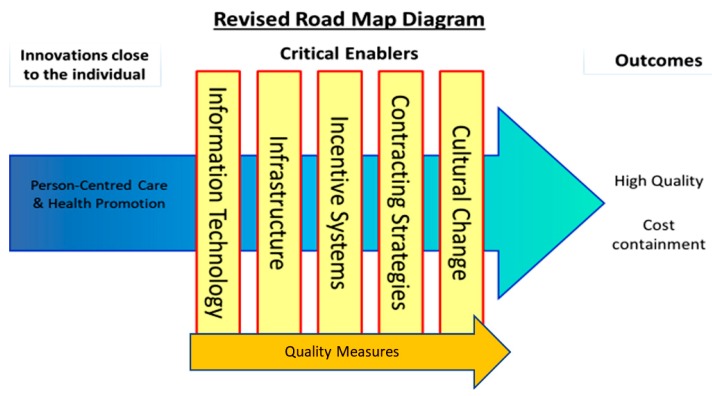
The intersections of critical enablers and the core process of person-centred care and health promotion.

**Figure 2 ijerph-17-03050-f002:**
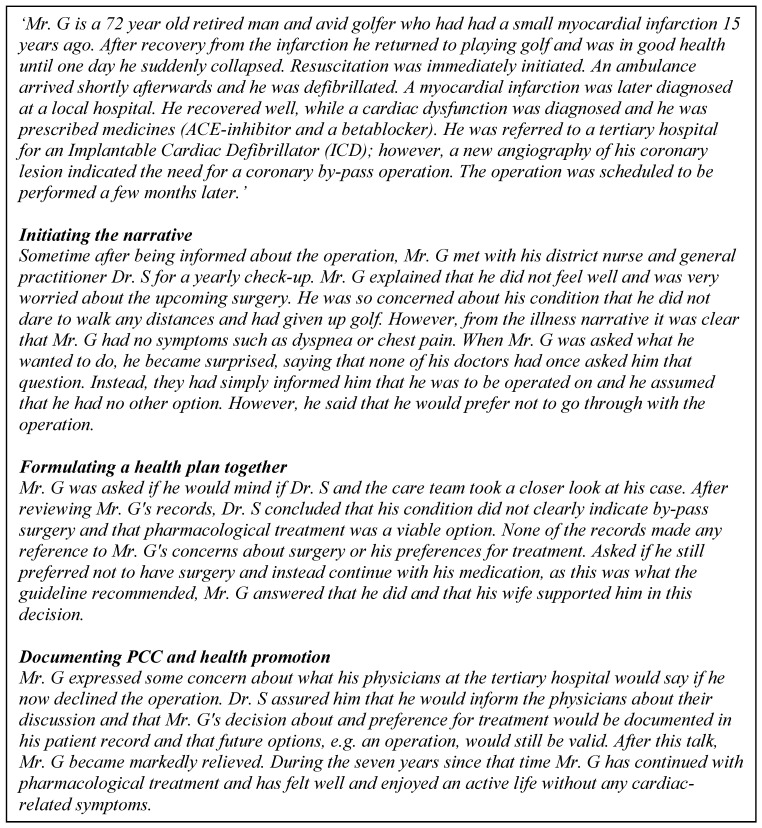
The illness story of Mr. G.

**Figure 3 ijerph-17-03050-f003:**
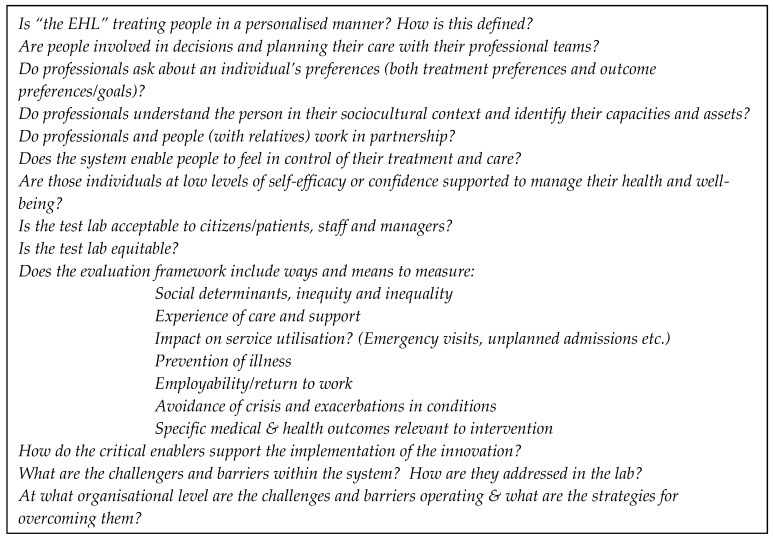
Questions for PCC practice and experience as a quality marker.

**Figure 4 ijerph-17-03050-f004:**
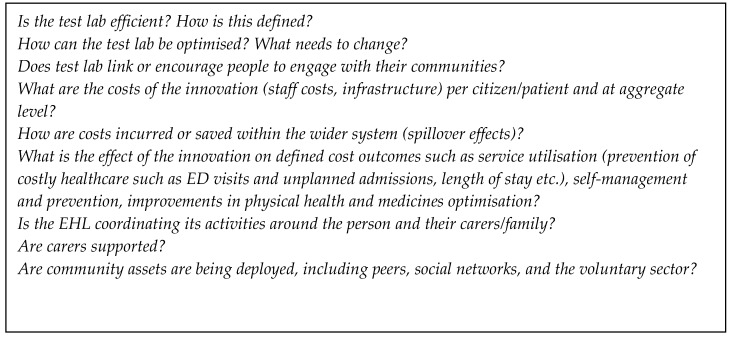
Questions for cost in the “health labs”.

**Table 1 ijerph-17-03050-t001:** Program theories, if-then-because statements, and measurement/assessment examples.

PT Type	IF-THEN	Because	Measurement/Assessment
A	IF delivery organizations, owners, financiers, and commissioners form an ‘Alliance’ contract to deliver the EHL based on shared and co-designed PCC/HP objectives, THEN the quality of PCC will improve, and costs may stabilize [42,43,44]	a context of trust based on the sharing of risk and reward is created across all parties with collective ownership of responsibilities of the EHL	Delivery team dynamics and communication. Aligned goals at macro, meso and micro levels. Progress against goals, e.g., number of PCC care plans with personal HP plan across providers. Performance of each provider towards the unified goal. Qualitative and quantitative measures (e.g., P3C-EQ, PPE15, P3C-OCT)
A	IF partnership models are created with community NGOs with PCC and HP agreed outcomes, THEN PCC, HP, and Cost containment will improve [45]	of increased access to social and HP activities, resulting in less reliance on medicines and treatments	Exploration of how outcomes were set, e.g., through stakeholder consultation, focus groups and interviews. Number HP activities. PCC processes. Patient experience of PCC (P3C-EQ)
B	IF incentives are provided at multiple levels, THEN this is more likely to lead to increased PCC [46]	it contributes to cultural change and the contingency is in place at micro, meso, and macro levels	Measure of cultural change (OR4KT) and organizational readiness for change (e.g., P3C-OCT). Interviews with stakeholders across levels
B	IF cost effectiveness of PCC and HP is measured on the whole PCC chain and savings divided between all participants, THEN all stakeholders will benefit	egoistic behavior of individuals in PCC delivery and organizational chains would be diminished	Comparative measurement of the cost of PCC provision/savings to all participants; include incentives and associated measurements at lower levels to avoid undesirable effects. Interviews with patients about their experience
C	IF contract payments are made at the same time to all partners and tied to PCC and HP outcomes, THEN this fosters trust and productivity towards aligned outcomes [47]	it helps to reduce misalignment/competition between partners and reduces transaction hazards	Monitor payment transactions (frequency and scheduling), organizational setting and structures, and how these affect transaction hazards. Interviews based on defined transaction hazards in alliance contract
D	IF incentives are provided to all individuals on the team (e.g., nurses, occupational therapists, etc.) and consist of the wider range of rewards (extrinsic and intrinsic rewards), THEN desired outcomes are more likely [48,49,50]	because everyone is on board to provide PCC and feels that the organization’s goals are congruent with their own	Patient experience questionnaires to measure PCC (P3c-EQ, PPE15). Objective evaluation from quality controllers. Professional and patient focus groups
E	If “Quality measures” are linked to PCC ideas and information systems (e.g., accounting system) and able to deliver information about cost containment or other quantitative indicators improvement against non-EHL settings (benchmarking), THEN the measurement process itself will be an incentive [51]	the measurement process has also the function of ex-ante control applied “Quality measures” enabler	Audit of use of quality measures and linkage to information system. Interviews with delivery and management staff on acceptability and effectiveness. Benchmarking with non-EHL settings
F	If mobile technologies (e.g., SMS reminders, mobile symptom monitoring etc.) are used by organizations, THEN PCC goals and HP are supported for self-management activities [52]	people are more receptive and in control of their own health	Number of interactions and activities performed by utilizing IT. Number of reviews. Number of follow-ups. Number of care contact changes over time. Patient surveys regarding experiences and outcomes following communication episodes. (PPE15. P3C-EQ)
F	IF IT systems are used to support dynamic and goal-oriented electronic health records (EHRs) that include patient inputs, THEN EHRs and care plans can be more PCC [53,54]	technology will enable patients to better manage their illness and care teams to address the patient’s overall needs, concerns, and goals with a single plan	Number of created functionalities to permit easy tracking of an individual patient over time (e.g., prior hospitalizations). Existence of care management software built into the EHR. Creation of huddle sheets and pre-visit planning tools that can be populated with important patient data (e.g., medications, problem list). Interviews with participants and care team
G	IF staff training in empowerment and communication is provided, THEN this improves PCC [55,56,57]	it enables relationships with patients as capable people, and both parties have greater participation in care	Number of staff trainings for PCC, empowerment workshops and reflection meetings. Outcomes measured by Individualized Care Scale, Ways of Coping Questionnaire and EQ-5D. Video recording and coding of healthcare provider—patient interactions
G	IF feedback is provided to patients from staff using data from patient-reported measures, THEN this can improve the responsiveness of clinicians and improve patient information and choice [58]	the patient perspective drives the care interaction and care plan	Sample of care plans compared with patient reported outcome measure (PROM) data. Interviews with staff and patients

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
