# Peer review of "Supporting Innovative Person-Centred Care in Financially Constrained Environments: The WE CARE Exploratory Health Laboratory Evaluation Strategy"

_ijerph, 2020, doi:10.3390/ijerph17093050_

Round 1

Reviewer 1 Report

Supporting Innovative Person-Centred Care in Financially Constrained Environments: the WE CARE Exploratory Health Laboratory Evaluation Strategy

Thank you for submitting your paper to the International Journal of Environmental Research and Public Health, and providing me with the opportunity to review your work. Interesting concept paper of the evaluation of the WE CARE study.

I only have a couple of questions:

  1. If the WE-CARE project was funded in 2013 and 2015, why is now a paper on an evaluation strategy, should this not have been developed and implemented alongside the study?
  2. When explaining IF THEN equations, PTs and IT needs to explained as these are not clear or apparent
  3. Within the evaluation strategy, what do you mean by the ‘true case-story’, even those these have been published the reader of this paper needs an understanding of these

Author Response

  • If the WE-CARE project was funded in 2013 and 2015, why is now a paper on an evaluation strategy, should this not have been developed and implemented alongside the study?

Thank you for your question. The WE-CARE project (2013-2015) was set up to bring together a consortium of experts to create a 'Road Map' as a detailed set of instructions to address quality and cost for modern health care systems in the EU. The Road Map was also a 'call to action' to to build the research and development expertise across the EU to deliver this. The COST CARES (2016-2020) project was set up to create the network and specific tools to do this. The COST CARES project is the subject of this paper.   

  • When explaining IF THEN equations, PTs and IT needs to explained as these are not clear or apparent

Thank you for raising this. I have inserted an explanation of Programme Theories on lines 105-111:

'Programme Theories are developed to describe how interventions (service, treatment, policy) are thought to work by specifying the ways in which they produce outcomes. PTs are also useful for understanding both the positive and negative impacts that can occur when interventions are implemented. They are often accompanied by logic models which help plan and evaluate interventions based on their internal logic, thus supporting successful delivery and evidence acquisition.'   

I have added this text to describe IT on lines 112-114:

1) Information technology (IT) which describes the use of computers or other computerized devises to store, transmit and receive data to support PCC planning and care coordination and for handling and communicating health and evaluation data, and for delivering PCC and HP interventions.

  • Within the evaluation strategy, what do you mean by the ‘true case-story’, even those these have been published the reader of this paper needs an understanding of these

I have added the text below on lines 160-161 to clarify what we mean by 'true case story'.

'The below true case story is a vignette based on a real person to illustrate how PCC can be applied in practice through a worked example.'

Reviewer 2 Report

The authors present a very interesting paper on patient centered care with what is essentially a population health perspective.

A few comments for the authors to respond to:

1) Line 35. Health Labs is introduced in the Abstract for the first time here. It is always difficult to extract full clarity in such situations, and it is unclear who "they" is referring to at the end of this line.

2) Line 48. It is unclear what global health crisis the authors are referring to. The references don't suggest any particular crisis either -- in fact several potential crises come to mind. Please clarify.

3) Line 90 and Line 215 and Figure 1. Patient centered care also involves ensuring that individuals feel that they are treated as individuals. Have the authors explored other forms of people first language? It is ok to refer to patients as patients within the confines of the health system and its infrastructure. But how do patients feel about being called patients? Do they prefer 'clients'? Should we be talking about "individuals with diabetes" rather than diabetics? How do we support public perception change such that advances in patient centered care supported and understood by the public and stakeholders? These are all questions that the authors would do well to comment on in the discussion section.

4) Table at Line 189 and following. This table is very good. But it lacks examples of specific measures. Whereas this might not be the authors' intent, it would help the manuscript.

5) This paper might be made more timeline with a discussion on virtual platforms for delivering and ensuring fidelity to PCC. May the authors consider this as a Discussion section point?

Author Response

1) Line 35. Health Labs is introduced in the Abstract for the first time here. It is always difficult to extract full clarity in such situations, and it is unclear who "they" is referring to at the end of this line.

I have removed the words 'as they' to avoid the confusion above.

2) Line 48. It is unclear what global health crisis the authors are referring to. The references don't suggest any particular crisis either -- in fact several potential crises come to mind. Please clarify.

The text on line 48 specifies 'global economic crisis', it is this crisis to which we refer. The references refer to this and the impact on health care see:

  1. Correia, T.; Dussault, G.; Pontes, C. The impact of the financial crisis on human resources for health policies in three southern-Europe countries. Health Policy (New. York). 2015, 119, 1600–1605.
  2. Pettoello-Mantovani, M.; Namazova-Baranova, L.; Ehrich, J. Integrating and rationalizing public healthcare services as a source of cost containment in times of economic crises. Ital. J. Pediatr. 2016, 42, 18.
  3. Yfantopoulos, N.; Yfantopoulos, P.; Yfantopoulos, J. Pharmaceutical policies under economic crisis: The Greek case. J. Heal. Policy Outcomes Res. 2016, 4–16.
  4. Ongaro, E.; Ferré, F.; Fattore, G. The fiscal crisis in the health sector: Patterns of cutback management across Europe. Health Policy (New. York). 2015, 119, 954–963.
  5. Wenzl, M.; Naci, H.; Mossialos, E. Health policy in times of austerity—a conceptual framework for evaluating effects of policy on efficiency and equity illustrated with examples from Europe since 2008. Health Policy (New. York). 2017, 121, 947–954.

3) Line 90 and Line 215 and Figure 1. Patient centered care also involves ensuring that individuals feel that they are treated as individuals. Have the authors explored other forms of people first language? It is ok to refer to patients as patients within the confines of the health system and its infrastructure. But how do patients feel about being called patients? Do they prefer 'clients'? Should we be talking about "individuals with diabetes" rather than diabetics? How do we support public perception change such that advances in patient centered care supported and understood by the public and stakeholders? These are all questions that the authors would do well to comment on in the discussion section.

We thank you for raising this issue, which highlights the importance of language. We do not have a discussion section but have on lines 86-91 provided a justification of why we favour the term 'Person Centred' over ‘Patient centred’. As our model and ethic applies to all persons irrespective of diagnosis, we also do not feel it necessary to discuss disease specific people first language. Similarly, we do not feel it necessary to discuss the term 'client centred' since this term is more representative of fee paying purchaser models of health care and our definition of Person Centred Care is intended to be applicable to all health care settings. Importantly, whilst we acknowledge the importance of language, we are too also cognisant of the length of this manuscript, the focus of which is the COST CARES strategy and not the philosophy or the language of PCC, which have been well described elsewhere. We hope that this is a fair response to this important issue.

4) Table at Line 189 and following. This table is very good. But it lacks examples of specific measures. Whereas this might not be the authors' intent, it would help the manuscript.

Thank you for raising this. We did have some measures specified e.g. see type g Ways of coping Questionnaire, Individualised care scale & EQ-5D. We have added examples of Patient and organisational measures to A, B, D and F. We do not have the space for a detailed list but refer readers to guidance and an online compendium to support decisions for measurement see line 279-280.

5) This paper might be made more timeline with a discussion on virtual platforms for delivering and ensuring fidelity to PCC. May the authors consider this as a Discussion section point?

I have added the following text on lines 265-268

'In the current situation, remote monitoring of patients, video-linked consultations and e-health interventions could provide an exciting opportunity to test the delivery of person-centered care remotely, with the potential to calculate costs compared to previous standard practice.'